# Ionising Radiation Induces Promoter DNA Hypomethylation and Perturbs Transcriptional Activity of Genes Involved in Morphogenesis during Gastrulation in Zebrafish

**DOI:** 10.3390/ijms21114014

**Published:** 2020-06-04

**Authors:** Sophia Murat El Houdigui, Christelle Adam-Guillermin, Olivier Armant

**Affiliations:** 1PSE-ENV/SRTE/LECO, Institut de Radioprotection et de Sûreté Nucléaire (IRSN), Cadarache, 13115 Saint-Paul-Lez-Durance, France; sophia.murat@gmail.com; 2PSE-SANTE/SDOS/LMDN, Institut de Radioprotection et de Sûreté Nucléaire (IRSN), Cadarache, 13115 Saint-Paul-Lez-Durance, France; christelle.adam-guillermin@irsn.fr

**Keywords:** AOP, development, ionising radiation, epigenetic, DNA methylation, transcriptomics, zebrafish, gastrulation, germ layer

## Abstract

Embryonic development is particularly vulnerable to stress and DNA damage, as mutations can accumulate through cell proliferation in a wide number of cells and organs. However, the biological effects of chronic exposure to ionising radiation (IR) at low and moderate dose rates (< 6 mGy/h) remain largely controversial, raising concerns for environmental protection. The present study focuses on the molecular effects of IR (0.005 to 50 mGy/h) on zebrafish embryos at the gastrula stage (6 hpf), at both the transcriptomics and epigenetics levels. Our results show that exposure to IR modifies the expression of genes involved in mitochondrial activity from 0.5 to 50 mGy/h. In addition, important developmental pathways, namely, the Notch, retinoic acid, BMP and Wnt signalling pathways, were altered at 5 and 50 mGy/h. Transcriptional changes of genes involved in the morphogenesis of the ectoderm and mesoderm were detected at all dose rates, but were prominent from 0.5 to 50 mGy/h. At the epigenetic level, exposure to IR induced a hypomethylation of DNA in the promoter of genes that colocalised with both H3K27me3 and H3Kme4 histone marks and correlated with changes in transcriptional activity. Finally, pathway enrichment analysis demonstrated that the DNA methylation changes occurred in the promoter of important developmental genes, including morphogenesis of the ectoderm and mesoderm. Together, these results show that the transcriptional program regulating morphogenesis in gastrulating embryos was modified at dose rates greater than or equal to 0.5 mGy/h, which might predict potential neurogenesis and somitogenesis defects observed at similar dose rates later in development.

## 1. Introduction

Chronic exposure to pollutions is associated in wild animals with immunosuppression, increased sensitivity to stress and cancerogenesis, all contributing to loss of fitness and raising concerns for species conservation [1,2,3,4,5,6,7]. Ionising radiation (IR) can induce double-strand DNA breaks that may lead to cellular senescence, cell death or cancer [8,9]. The early phases of embryonic development are particularly sensitive to stress and DNA mutations [10], as somatic mutations can accumulate through cell proliferation in a wide number of cells and organs, increasing the risks of morphological defects and cancers [11], especially at high doses of IR (more than 0.1 Gy). However, the biological effects of low dose rates of IR (defined by UNSCEAR at < 6 mGy/h) [12] remain largely unknown.

Tight control of transcriptional activity is necessary to orchestrate the dynamic processes of embryonic development. Epigenetic modifications like DNA methylation, histone modifications and noncoding RNAs, are important modulators of gene activity during development. Just after fertilisation, the zygote divides rapidly during the cleavage stage, relying on the maternally deposited material (mRNA and proteins) for cell division, until it reaches the maternal to zygotic transition (MZT) [13,14]. The level of DNA methylation changes dramatically during this early phase of embryogenesis, as the parental DNA methylation marks are erased rapidly in order to reach totipotency [15]. This process of DNA methylation reprogramming is observed in rodents and zebrafish, which suggests that it could be a common process in vertebrates, but differences exists between these species. In mice, erasure of DNA methylation marks is almost complete and few loci escape, like imprinted genes. In contrast, studies in zebrafish demonstrated that DNA methylation marks are not extensively erased but are rather decreased just after fertilisation [16,17]. After the MZT, the diploid embryonic genome becomes activated and starts to produce mRNA, which requires the epigenetic marks, including DNA methylation patterns, to be re-established in order to proceed with morphogenesis. At gastrulation, the morphogenetic mechanism of convergence and extension shapes the rudimentary body plan, together with the formation of the three germ layers, ectoderm, mesoderm and endoderm, at the origin of all tissues and organs in the organism. The gene regulatory networks that drive vertebrate germ layer morphogenesis are well documented, especially in zebrafish and in *Xenopus* as these species are easily amenable to experimentations after fertilisation [18,19], which allows a detailed analysis of altered processes.

Changes in epigenetic marks have already been observed after exposure to high and low doses of IR (lower than 0.1 Gy). At high doses (5 Gy), the expression levels of DNMTs and methyl CpG binding proteins (MeCP2) were decreased, leading to global DNA hypomethylation [20], a sign of genomic instability [21] and one of the first epigenetic abnormality discovered in cancer cells [22]. However, fewer studies are available in the field of low doses. For instance, exposure of pregnant mouse from 0.7 to 7.6 cGy resulted into dose- and sex-dependent epigenetic modifications at the A^vy^ locus in the offspring [23]. In another study, exposure of zebrafish embryos to IR at 10.9 mGy/h for 3 h modified the H3K4me3 histone mark in important developmental genes like *vegfab*, *geminin* and *hnf4a*, but effects on DNA methylation were not investigated [24]. Altogether, these results suggest that IR can induce epigenetic changes during embryogenesis. However, how these epigenetic changes affect gene expression, with possible harmful consequences on embryonic development, remains largely unknown.

We previously demonstrated that IR at 0.5, 5 and 50 mGy/h alter the transcriptional program of neurogenesis and somitogenesis in 24 hpf zebrafish embryos, and that these molecular perturbations correlated with decreased larval motility at 120 hpf [25]. The present study focuses on the transcriptomic and epigenetic effects of IR on gastrulation. Increasing dose rates of gamma IR spanning five orders of magnitude, from 0.005 to 50 mGy/h, were used. The two lower dose rates, 0.005 and 0.05 mGy/h, encompass the benchmark dose rate of 0.01 mGy/h recommended for ecosystem radiological protection [26] and were thus defined as low dose rates. The dose rates of 0.5 and 5 mGy/h, in the range of the DCRL [27] where phenotypical effects start to be observed in juvenile fish, were defined here as moderate. Finally, we defined 50 mGy/h, typical of exposures shortly after the Chernobyl accident, as a high dose rate. A system biology approach was chosen to study the epigenetic modifications (by whole genome bisulphite sequencing, WGBS) and changes in transcriptional activities (by RNA-seq) induced by IR during gastrulation (6 h post-fertilisation, hpf). This stage of development corresponds to the shield stage, roughly at the middle of the gastrulation, when the three embryonic germ layers (mesoderm, endoderm and ectoderm) are being set up [28]. The effects of IR on gene expression were investigated by RNA-seq at all dose rates (from 0.005 to 50 mGy/h). DNA methylation changes were studied at 5 and 50 mGy/h, two dose rates at which we anticipated more detectable effects.

## 2. Results

### 2.1. A Transcriptome-Wide Analysis Reveals Changes in the Expression of Genes Involved in Morphogenesis after Exposure to IR

A transcriptomic analysis was performed in order to identify the biological pathways altered by IR during the early phase of zebrafish development. Fertilised embryos were exposed continuously to IR until the shield stage (6 hpf) at dose rates of 0.005, 0.05, 0.5, 5 and 50 mGy/h. A total of 37 samples including controls (at least 3 replicates per condition) was used for pairwise differential expression analysis (Appendix A). No obvious morphological changes or increased mortality were observed at any of the dose rates tested here. The number of differentially expressed genes (DEG) within our significance threshold (|fold change| ≥ 1.5 and adjusted *p*-value < 0.01) increased nonlinearly with the dose rate: 39 DEG at 0.005 mGy/h, 25 DEG at 0.05 mGy/h, 654 DEG at 0.5 mGy/h, 668 DEG at 5 and 2637 DEG at 50 mGy/h. No DEG was common to the five dose rates (Figure 1a). The largest overlap was observed for the two highest dose rates, 5 and 50 mGy/h (589 DEG, 21.6%, *n* = 2716 genes), which suggests that the transcriptomic responses induced in these two conditions were partially overlapping. The three embryonic layers, ectoderm, mesoderm and endoderm, are formed during gastrulation. After exposure to IR, genes involved in the specification of the three germ layers were impacted at all dose rates tested here (Figure 1b). Well-known transcription factors (TF) involved in morphogenesis, like members of the *fox*, *tbx*, *cdx*, *lft*, *lhx*, *sox*, *bmp* or *gata* families, were misregulated. Two different clusters were identified based on hierarchical clustering of gene expression. Cluster 1 (red) was composed of genes upregulated at 5 and 50 mGy/h but also, to a lower extend, at the other dose rates, while cluster 2 (green) was composed of genes mostly downregulated in the different conditions. Exposure to IR can thus alter the expression of genes important for germ layer morphogenesis at 5 and 50 mGy/h, but also at low dose rates below 0.5 mGy/h.

We then checked if some genes displayed a dose-dependent response to radiation by analysing the different expression patterns of 3319 DEG (see Material and Methods). Four different patterns of expression were discriminated in response to increasing dose rates of radiations (Figure 1c). Two clusters were composed of genes that responded nonlinearly to the dose rates. Cluster 1 (*n* = 750 genes) was characterised by genes strongly upregulated only at 0.5 mGy/h, while genes in cluster 2 (*n* = 794 genes) were downregulated at 0.5 mGy/h but upregulated at 50 mGy/h (Figure 1c,d). In contrast, genes in clusters 3 (*n* = 459 genes) and cluster 4 (*n* = 927 genes) displayed a linear response to the dose rate, the genes being increasingly upregulated or downregulated from 0.5 to 50 mGy/h (Figure 1c,d). These results indicate that a part of the transcriptional response to IR follows a dose-dependent response.

### 2.2. Moderate and High Dose Rates of IR Impact Biological Pathways Involved in Ectoderm and Mesoderm Development

A pathway enrichment analysis was performed using the Gene Ontology (GO) and KEGG (Kyoto Encyclopedia of Genes and Genomes) repositories. Key developmental processes such as regulation of mesodermal fate (erythrocyte, myeloid cell differentiation and somitogenesis), ectodermal fate (nervous system development) as well as canonical Wnt signalling were impacted at the two highest dose rates (Figure 2, Appendix A). In addition, several genes of the retinoic acid pathway (RA) involved in the regionalisation of the three germ layers along the anteroposterior axis (*crabp2b*, *rarab*, *aldh1a2*) were upregulated at 5 and 50 mGy/h (Appendix A). Fewer biological pathways were enriched in the other conditions (from 0.005 to 0.5 mGy/h) due to the small number of DEG. However, a significant enrichment of genes involved in mitochondrion activity (electron transport chain, oxidative phosphorylation and ATP synthesis-coupled electron transport) was found at 0.5 mGy/h and higher dose rates (Appendix A). As IR exposure leads to partially overlapping transcriptomic response at 5 and 50 mGy/h (21.6%, 589 DEG), we checked the common deregulated pathways in these two conditions. A significant enrichment of pathways involved in embryonic development (pattern specification process and canonical Wnt signalling pathway), cell differentiation (polarised epithelial cell differentiation), mesoderm development (somitogenesis) and metabolism (regulation of lipid biosynthetic process and peptide metabolic process) was found (Appendix A).

### 2.3. Promoter Analysis Reveals Enrichment of Master Regulators Expressed in the Three Germ Layers and Involved in Mitochondria Energetic Metabolism

We analysed the promoter (genomic sequence 2 kb upstream and 50 bp downstream the transcriptional start site) of all DEG to identify master transcriptional regulators that orchestrate gene expression, focusing on the three highest dose rates as such analysis requires large number of DEG. This analysis allows the characterisation of the upstream transcription factors potentially responsible for the observed transcriptional changes in the RNA-seq data. We found 84 different TF DNA-binding sites enriched in the promoter of the DEG at 0.5 mGy/h, 51 at 5 mGy/h and 80 at 50 mGy/h (Appendix A). By checking the expression of these TF during zebrafish embryogenesis, we found that 24% were expressed during gastrulation (Appendix A). We then analysed the DNA-binding sites in common between the three dose rates and found an enrichment of TF involved in the development of the three germ layers (FOXG1, NKX6-1 and POU3F3) as well as TF regulating glycolysis in mitochondria (FOXK1 and FOXK2) [29] (Appendix A). These data show that TF regulating morphogenesis or involved in energetic metabolism were altered by exposure to IR at dose rates greater or equal to 0.5 mGy/h.

### 2.4. Embryonic Exposure to Moderate and High Dose Rates of IR Induces Promoter Hypomethylation

To assess whether IR can modify epigenetic marks and thereby modulate transcriptional activity, we analysed DNA methylation at the whole genome scale using WGBS on shield stage embryos. The two highest dose rates of 5 and 50 mGy/h were used as these exposures lead to many DEG. The methylation levels of 18,322,470 and 17,759,785 CpG were quantified, respectively, at 5 and 50 mGy/h, and compared to controls. The global methylation level of 5-methyl cytosine (5mC) in the CpG context was close to 80% in the control group (Figure 3a), in accordance to previous zebrafish studies on similar embryonic stages [16]. No change in global methylation levels was detected after exposure to IR (Figure 3a). As expected, a fraction of known CpG islands (CGIs) was highly methylated in control embryos and confirmed the quality of the WGBS data (Figure 3b). The pairwise differential analysis of CpG methylation levels between control and exposed conditions highlighted 8190 and 7560 differentially methylated cytosines, respectively, at 5 and 50 mGy/h (adjusted *p*-value < 0.05 and methylation difference ≥ 10%). To check if exposure to IR can modify methylation levels in CGIs, methylation levels between the control and exposed embryos at 5 and 50 mGy/h were compared. Both hypo- and hypermethylation of CGIs were detected, but hypomethylated CGIs were predominant in both conditions (Figure 3c,d). As CpG are usually used as a proxy for promoter localisation, we checked directly the methylation status of known promoters. As for CGIs, a fraction of known promoters was highly methylated in the zebrafish genome (Figure 3e) and promoter hypomethylation was predominant after exposure to IR at 5 and 50 mGy/h (Figure 3f,g). It is usually assumed that promoter hypomethylation is associated with an open chromatin state that activates transcription. To check whether IR-induced DNA hypomethylation was associated with an active or inactive transcriptional state, we mapped the 5mC obtained by WGBS against the histone marks H3K27me3 and H3K4me3 from a published ChIP-Seq dataset made at the same developmental stage [30], as these marks are associated with promoters repression and activation, respectively, after the MZT [31]. High levels of 5mC were observed in the vicinity of both histone marks in controlled conditions, as expected (Figure 3h,k). Furthermore, DNA hypomethylation was observed in the vicinity of both H3K27me3 (Figure 3i,j) and H3K4me3 (Figure 3l,m) marks, showing that DNA hypomethylation could be associated with both repressed and active promoters. High doses of IR are known to induce genomic instabilities and can be associated with the mobilisation of transposons. To check if such processes might occur in our exposure scenario, the DNA methylation level of the different transposon families present in zebrafish [32] was checked. No change in DNA methylation was detected in DNA transposons or retrotransposons (Appendix A).

To further assess how DNA methylation is impacted after exposure to IR, we grouped 5mC into differentially methylated regions (DMR), a common way to study DNA methylation differences across biological samples and assess their possible function in transcriptional regulation [33] (see Material and Methods). We found 1858 and 1208 DMR (methylation difference > 10%, permutation *p*-value < 0.01) in the zebrafish genome after exposure to 5 and 50 mGy/h, respectively. Among these, 297 DMR (either hyper- or hypomethylated) were found in common at both dose rates and separated by less than 2 kb. The comparison of their methylation status at 5 and 50 mGy/h showed that DMR were mostly hypomethylated compared to controls, and 60% (corresponding to 177 DMR) were differentially methylated concordantly at 5 and 50 mGy/h (Figure 4a). DMR localised in promoter regions are classically considered to modulate gene expression in vertebrates. We, therefore, studied more closely the distribution of DMR around the different genomic feature (promoters, introns, exons, UTRs and intergenic regions). Most DMR were located in intergenic regions or introns but up to 20% of the DMR were located at less than 3 kb from known promoters (Figure 4e). Focusing on DMR located close to promoters, we found that DMR were enriched at the TSS in both exposed groups (Figure 4b). In addition, the discrimination between hypo- and hypermethylation showed that DMR localised in the TSS region were hypomethylated at both dose rates (Figure 4c,d). Taken together, these results show that exposure to IR leads to a reduction of DNA methylation in gene promoter and indicate a potential functional role on the modulation of transcriptional activity.

### 2.5. Promoter Hypomethylation after Exposure to Moderate and High Dose Rates of IR Modulates Transcriptional Activity of Important Developmental Genes

To check if promoter hypomethylation can have an impact on gene activity, we crossed the WGBS data with the expression analysis made by RNA-seq. A total of 600 hypomethylated DMR, located at less than 500 bp of known TSS and hypomethylated at 5 or 50 mGy/h, were selected, and the expression of the corresponding genes was analysed by hierarchical clustering (Figure 5). We observed two different clusters of genes: cluster 1 (red) was composed of genes expressed at low level in controls and upregulated after irradiation at 5 and 50 mGy/h, while cluster 2 (green) was composed of genes that were mostly downregulated in the exposed conditions compared to controls. DMR hypomethylation was thus correlated with a modulation of transcriptional activity, but not strictly with the upregulation of gene expression. We then checked if we could detect differentially methylated promoters (hyper- or hypomethylated) associated with significant transcriptional changes (DEG with adjusted *p*-value < 0.05). We detected 227 DEG with DMR at 50 mGy/h (Figure 6, two first outer circles), containing a differentially methylated promoter, and 99 genes at 5 mGy/h (Figure 6, two inner circles). For instance, *twist1a* expressed in mesoderm during gastrulation [34] was significantly upregulated eight times (adjusted *p*-value < 10^−7^) in zebrafish embryos exposed at 50 mGy/h compared to controls, and displayed a 10% hypomethylation in a DMR located 804 bp upstream *twist1a* TSS (Figure 7a, upper panel). Another gene, *blf*, expressed in mesoderm and involved in convergent extension during gastrulation [35], contained a hypermethylated DMR (29%) located 2258 bp upstream TSS and was downregulated 1.8-fold in embryos exposed at 50 mGy/h (adjusted *p*-value < 10^−2^) (Figure 7a, lower panel). At 5 mGy/h, *fgf4* implicated in left–right symmetry was downregulated 2.6-fold and a hypomethylated (10%) DMR was detected at 205 bp from its TSS (Figure 7b, upper panel). Similarly, a DMR located in the *bmp2b* promoter (distance to the TSS = 0), a morphogen involved in specification of the ventral fate in gastrulation [36], was hypomethylated at 5 mGy/h and associated with transcriptional activation (adjusted *p*-value < 0.05 and fold change = 1.7). These results confirm that DMR located in important genes regulating morphogenesis can be associated with significant transcriptional changes, but also highlight that, in our data, promoter hypomethylation was not strictly associated with transcriptional activation.

Finally, a functional annotation of the differentially methylated promoter was made using DMR located at less than 3 kb from known TSS. The GO pathways associated with promoter methylation changes were involved in general process of embryonic development (developmental growth, specification of symmetry and regulation of gastrulation), ectoderm development (telencephalon development, hindbrain development, spinal cord development, regulation of neuron development, synaptic signalling and neurogenesis) and mesoderm development (somitogenesis) (Figure 8).

## 3. Discussion

In this study, we investigated how the early step of zebrafish embryonic development can be altered after exposure to IR by studying genome-wide gene expression and DNA methylation. No morphological abnormalities or increased embryonic lethality could be observed at any of the dose rates tested here. This is in agreement with earlier studies that showed that IR exposure at less than 150 mGy does not impact embryonic survival directly [37], but rather induces subtle neuromuscular and motility defects [25]. However, we observed different molecular effects depending on the dose rates, from modification of mitochondrial processes at moderate and high dose rates (> 0.5 mGy/h), to perturbation of important TF involved in morphogenesis of ectoderm and mesoderm at the two highest dose rates (5 and 50 mGy/h). The transcriptional profile obtained at 5 mGy/h was largely overlapping the one at 50 mGy/h, and, as expected, dose rates at 0.5 mGy/h and below induced subtle changes on gene expression. However, a high number of unique genes were found at 50 and 0.5 mGy/h indicating that a particular transcriptional response might occur at these two dose rates. In addition, the expression of several genes varied proportionally with dose rates from 0.5 to 50 mGy/h, while others displayed nonlinear patterns especially around 0.5 mGy/h. These results suggest that, in our study, a linear dose–response with irradiation is not a common feature to all genes and all dose rates. Rather, we observe that a specific set of genes responded linearly with dose rates between 0.5 and 50 mGy/h, but not at lower dose rates. Similar nonlinear response to gamma radiation has already been described in mouse and plants for cytogenetic damages like micronuclei and chromosomic aberrations [38].

Functional analysis made on the transcriptomic data showed a significant enrichment of genetic pathways involved in neuroectoderm and mesoderm development at 5 and 50 mGy/h. More specifically, molecular processes regulating neurogenesis (GO:0060322: head development, GO:0007399: nervous system development and GO:0030902: hindbrain development), somitogenesis (GO:0061053: somite development) and differentiation of blood cells (GO:0061515: myeloid cell differentiation and GO:0043249: erythrocyte differentiation) were significantly enriched at the two highest dose rates. However, these organs are not formed yet in shield embryos (6 hpf). The detailed analysis of the deregulated genes present in these GO terms, highlights the presence of many genes involved in Notch signalling (*deltaB*, *her6*, *her12* and *notch3*), as well as morphogens (*shha* and *bmp2b*) and TF (*sox2*, *six3b*, *otx2a* and *lft1*). All these genes are known to be expressed during gastrulation but are also involved in organogenesis later in development (reviewed in [39,40,41]). We thus interpreted these results as a deregulation of morphogenesis in the corresponding germ layers, neuroectoderm and mesoderm for the central nervous system and somite development, respectively. The deregulation of TF specifically expressed in the ectoderm (*msx1a*, *otx2b*, *lft2* and *her2*) and in the mesoderm (*tbx16l*, *fzd2*, *tbx6* and *twist1a*), during gastrulation, support the hypothesis that ectoderm and mesoderm were impacted at the molecular level by IR. How these molecular effects affect gastrulation or neurogenesis and somitogenesis at the cellular level was not investigated in the present study. And further studies are required to confirm the observed transcriptional changes at other levels. However, it was observed in our previous study (using the same experimental settings) that IR at 5 and 50 mGy/h altered the expression of genes involved in neurogenesis and somitogenesis in 24, 48 and 96 hpf zebrafish embryos and larvae, and these molecular effects were linked to neuromuscular impairments and larval motility defects in 5 days old larvae [25]. From these complementary sets of data, it can be proposed that IR at 5 and 50 mGy/h deregulates gastrula stage neuroectoderm and mesoderm morphogenesis, which might lead later in development to central nervous system and muscle impairments. Another study demonstrated neuromuscular impairment as well as a decrease of acetylcholinesterase expression during chronic exposure at similar dose rates, reinforcing prior results showing that IR at more than 5 mGy/h can lead to neuromuscular disorders [42]. All these results indicate that the developing central nervous system seems particularly vulnerable to stress and DNA damage [11,43]. In line with this observation, epidemiological studies conducted on Hiroshima and Nagasaki survivors demonstrated brain development defects and reduced cognitive performance in foetus exposed in uterus at doses higher than 0.31 Gy [44,45,46]. Likewise, recent field studies showed similar effects on wild animals, as decreased brain and body size were reported in monkey foetuses obtained in Fukushima Prefecture [47] and brain size of young birds living in the Chernobyl exclusion zone were also found to be smaller than in control area [48]. Taken together, our results indicate that the functional defects in neurogenesis and muscle development observed at 24 hpf and up to 5 days post fertilisation, might have their root, at least in part, in early developmental perturbations in the morphogenesis of the neuroectoderm and the mesoderm.

In a recent study, Hurem et al. [49] described the effects of IR (at dose rates higher than 0.54 mGy/h) on zebrafish gastrulation. The authors described a deregulation of RA and Notch signalling, as well as other important developmental genes like *vegfab*, *apoA1b*, *sox2* and *vox*. In addition, the authors used the Ingenuity software (QIAGEN, Inc., California, USA https://targetexplorer.ingenuity.com/) to describe potential upstream regulators (*myc*, *tp53*, *tnf* and *hnf4a*) and potential disease networks (brain malformation, growth failure, necrosis and hypoplasia of organs). If the effect on important developmental pathways (in particular RA and Notch signalling) was clear, the authors did not describe how IR can alter the morphogenesis of the three germ layers. In addition, the relationship between upstream regulators and downstream effects did not reveal any clear hypothesis on how the TF expressed during gastrulation were altered and how these effects could translate to detrimental effects at later stages (besides the potential role of *myc* in the induction of tumorigenesis and t*p53* induction of apoptosis). Presumably, the usage of the Ingenuity software precluded such analysis, as its knowledge-base is using annotations from human, rat and mouse [50] but not zebrafish. In the present study, we confirmed the deregulation of Notch and RA signalling at 5 and 50 mGy/h. Furthermore, we found that other pathways such as Wnt and BMP signalling were also altered. The identification of DNA-binding site of key TF expressed in ectoderm and the mesoderm in the DEG promoter consolidates the hypothesis that exposure to IR at dose rates higher than 5 mGy/h deregulates morphogenesis during gastrulation, with potential harmful consequences on neurogenesis and somitogenesis later in development.

In addition to the effects on the transcriptome, our WBGS results demonstrated that embryonic exposure to IR at 5 and 50 mGy/h can alter DNA methylation patterns outside, but also in the vicinity of TSS. Between 20% and 25% of the DMR were located in promoter regions (distance < 3 kb from TSS), while 60% to 70% of the DMR were located inside intronic or intergenic regions. Importantly, the functional annotation of differentially methylated promoters (DMR located at < 3kb from TSS) showed an enrichment of genes involved in embryonic development, neurogenesis and somitogenesis. These results correlate nicely with the transcriptomic data and collectively point towards a deregulation of genes involved in the morphogenesis of ectoderm and mesoderm upon exposure to 5 and 50 mGy/h. Mechanistically, these data suggest that IR could lead to changes in promoter methylation at the origin of the modulation of gene activity we observed at the transcriptomics level. It was already shown that exposure to high doses of IR decreases the expression levels of DNMTs and MeCP2 leading to global DNA hypomethylation [20,51]. If such mechanisms occur during embryonic development for low doses of IR remains to be confirmed, but no changes in DNMT expression was detected in our data (data not shown). How IR induces promoter hypomethylation during embryogenesis remains to be deciphered, as well as whether these methylation changes modify gene activity directly or not.

The distinction between hypo- and hypermethylated regions showed that most DNA methylation changes in promoters and in CGIs corresponded to a hypomethylation of these DNA regions. Previous studies demonstrated that demethylation of promoters positively modulate transcription [52]. By crossing the WGBS data with the expression data obtained by RNA-seq, we detected that hypomethylated DMR located at less than 500 bp from TSS were linked with transcriptional activation in about half of the cases, the other half of the genes being downregulated after exposure compared to controls. In addition, both H3K4me3 activating and H3K27me3 repressing marks were found in the vicinity of the hypomethylated DMR. Even if developmental genes can be poised for activation and thus harbour both H3K4me3 and H3K27me3 marks [53], these results show that DMR can be located in functional gene promoter. But establishing a clear link between DNA hypomethylation and gene activation remains difficult, as other factors (histone code, nucleosome assembly and TF-binding sites occupancy) affect also the chromatin state. Thus, the interaction of DMR with histone marks remains to be determined, for instance, by producing ChIP against H3K4me3 and H3K27me3 on irradiated embryos. In addition, functional studies on particular target genes will help deciphering the causal relationship between methylation changes, transcriptional regulation and adverse outcomes.

Hypomethylation of DNA has already been observed in previous studies after acute exposure to high dose of IR (0.5 to 5 Gy) either in cell lines [54] or in rodents [55,56,57]. Global DNA hypomethylation can lead to genomic instability [58,59] and the reactivation of transposable elements. We did not observe any changes in global methylation levels, nor in LINE elements or other transposable elements in the zebrafish genome, which suggest that the highest total dose rate used in this study (50 mGy/h during 6 h, i.e., total dose of 300 mGy) is not sufficient to cause genomic instability. Rather, our results indicate that the DMR associated with a role in gene regulation could be part of the stress response induced by IR. A recent study on zebrafish analysed the effects of a 27 days parental exposure at 8.7 mGy/h on DNA methylation changes in nonexposed F1 embryos at 50% of epiboly (5.5 hpf) [60]. Kamstra et al. found 5658 DMRs, predominantly located at regulatory regions (promoters and enhancers), and did not observe differences in the number of hypo- and hypermethylated DMR. As a comparison, we found 1858 and 1208 DMRs after exposure to 5 and 50 mGy/h, respectively. Despite differences in the exposure scenario (parental compared to direct exposure of embryos), these results suggest that longer exposure to IR (27 days compared to the 6 h used in our study) induce more effect on DNA methylation. DMRs and DEGs analyses on the F1-derived progeny pointed to an alteration of axonal guidance signalling. If axonal guidance is not a biological process ongoing in gastrulating embryos, these results can suggest an impairment of neurogenesis later in development (as proposed by the authors), which is in accordance with our present study where fertilised eggs were directly exposed to IR. These results raise the possibility that modification of DNA methylation patterns induced by IR can occur more frequently in the vicinity of developmental genes that could constitute IR-sensitive hotspots.

Our transcriptomic analysis detected potential effects of IR on mitochondrial energetic metabolism at dose rates higher than 0.5 mGy/h, indicating that this organelle could be sensitive to IR. Previous studies detected effects on mitochondrial activity for high doses (> 0.1 Gy) [61]. Our data show that mitochondrial activity could be altered at low dose rates (< 6 mGy/h). The fact that mitochondria possess less efficient DNA repair mechanisms compared to the nuclear genome [62] could, at least in part, explain our observation. Interestingly, mitigation of oxidative stress and DNA methylation share one biomolecule in their biological pathway, i.e., the S-adenosylmethionine (SAM). Indeed, the intracellular oxidative stress can be reduced in the cells by the synthesis of glutathione, a potent antioxidative molecule [63]. The limiting substrate for glutathione biosynthesis is the cysteine, which is itself synthesised from methionine via transsulphuration [64]. During glutathione synthesis, the methionine is converted into cysteine through reactions that involve SAM. As SAM is also the substrate used by DNMT enzyme to methylate DNA, a competition between DNA methylation and glutathione synthesis can occur in case of limited SAM bioavailability [65]. Thus, the redox status in the cell has an influence on SAM usage and may impact DNA methylation [66]. IR is known to increase the production of reactive oxygen species (ROS) [42] and produce DNA damages even in early gastrula embryos [67]. In the case of prolonged exposure to oxidative stress, glutathione stock can be depleted to protect cell from oxidative stress, which can result in an impairment of DNA methylation [68,69,70]. In our study, we found an impact on genes involved in oxidative stress and in mitochondrial redox processes at all dose rates but especially at dose rates higher than or equal to 5 mGy/h, suggesting that the redox balance during embryogenesis can be modified during exposure to IR. We found, for instance, that *gpx4a* and *gbx4b* expression, involved in the reduction of hydrogen peroxide, is increased at 5 and 50 mGy/h, as well as *duox*, *bco2l* and *nox1* at 50 mGy/h, which are involved in oxidative stress protection. It is thus possible that the increases of oxidative stress after exposure to IR can lead to partial depletion the SAM stock, at the expense of DNA methylation. However, such scenario would lead to global DNA hypomethylation of the genome, which is not observed in our data, but for much higher doses of IR [54,55,56,57,71]. More likely, our data points towards changes in mitochondrial activity as part of the stress response induced by IR, but more data on mitochondrial activity after exposure to low/moderate dose rates of IR (> 0.5 mGy/h) are needed to answer this question.

Our study explored the effects of low to high dose of IR on early embryonic development of zebrafish. Promoter hypomethylation at 5 and 50 mGy/h was associated with significant modulation on gene expression highlighting changes in the expression of gene involved in germ layer development. This observation underlines the role of the epigenetic mechanisms in the understanding of the effects caused by IR. Concordantly, many important developmental pathways like RA, BMP and Wnt signalling were impacted at the transcriptional level at 5 and 50 mGy/h, while few effects were detected at lower dose rates (0.5, 0.05 and 0.005 mGy/h). The transition between the effects induced by low and high dose rates seems thus to be located between 0.5 and 5 mGy/h during embryogenesis (total dose between 3 and 30 mGy, respectively). This is less than the proposed low dose and low dose rate limits defined so far in human (100 and 6 mGy/h [12]), but in the range of the expected dose rates giving rise to observable effects in the fish (0.4 to 4 mGy/h). In addition, we showed that transcription of genes involved in mitochondrial physiology was impacted at dose rates > 0.5 mGy/h. Taken together, our data suggest that the early developmental perturbations in the morphogenesis of the neuroectoderm and the mesoderm might predict the functional defects in neurogenesis and muscle development observed at later stages. From these different data, we propose a model of adverse outcome pathway of IR on embryonic development where perturbations of germ layer morphogenesis during gastrulation can contribute to the neurological and muscle disorders observed at later developmental stages.

## 4. Methods

### 4.1. Animal Experimentation and Ethics

Animal care was performed as described before [25]. Briefly, 6–9 month-old adults wild-type zebrafish (*Danio rerio*) of the AB strain (Amagen, Gif-sur-Yvette, France) were maintained in a ZebTEC system (Techniplast, Decine Charpieu, France) at 28 ± 1 °C, 350–450 μS/cm, pH 7.5 and a photoperiod of 12/12 h. In particular, 20 fishes were housed per 8 L tank. Animals were fed three times per day with TetraMin food flakes (Tetra Werke, Germany). Health was monitored by daily inspection. Embryos were obtained by mating 2 males and 2 females in 1.7 L breeding tanks for 15 min in fresh water (Beach Style Design, Techniplast, France) at a temperature of 28 ± 1 °C. Eggs from all spawn were pooled and grown in 25 mL embryo medium (60 mg/L Instant Ocean, 0.01% (w/v) of methylene blue) under constant temperature (28.5 ± 0.2 °C) and dark light cycles in MIR-154 incubator (Panasonic). Viable embryos were grown up to 6 hpf. All experiments were conducted with a percentage of fertilisation > 80%.

### 4.2. Irradiation

Gamma ray irradiations were performed in the MICADO experimental irradiation facility (IRSN, Cadarache, France) with four ^137^Cs sources of 370 GBq (Framatome ANP, Pierrelatte, France). The background level in the installation outside the irradiator was measured by operational dosimetry and was always <0.1 µGy/h. Fertilised embryos were exposed to gamma irradiation from 1 hpf (four-cell stage) up to 6 hpf. Absorbed dose rates in 100 mm diameter petri dish containing 25 mL of fish water and air kerma rates were computed with the Monte-Carlo N-Particle transport code (MCNPX version X-24E). Operational dosimetry with radiophotoluminescent dosimeters (RPL, GD-301 type, Chiyoda Technol Corporation, Tokyo Japan) was used to confirm the simulations at 50, 5, 0.5, 0.05 and 0.005 mGy/h. The effective dose rates were 46.80 ± 0.98, 5.08 ± 0.02, 0.47 ± 0.005, 0.052 ± 0.001 and 0.0044 ± 0.0001 mGy/h (Appendix A).

### 4.3. Sequencing Libraries Preparation and Data Generation for Transcriptomic Analysis

Total RNA extraction of biological replicates (3 to 6 per condition) were made from pools of 40 embryos at 6 hpf. Total RNA extraction was performed by TRIzol/chloroform extraction (Life Technologies). RNA integrity (RIN), quality and concentration were assessed using RNA Nano Chips (Bioanalyzer 2100, Agilent). All samples had a RIN > 8. Sequencing libraries were generated from 1 µg of total RNA following the TruSeq mRNA stranded protocol (Illumina, San Diego, USA). After quality check and concentration determination on DNA1000 Chips (Bioanalyzer 2100, Agilent, Les Ulis, France), libraries were run on a NovaSeq 6000 platform to produce 50 bases long paired-end reads (Clinical Research Sequencing Platform, Broad Institute, MIT, Cambridge, USA). Between 27 and 158 million of good-quality reads (Q > 30) were produced for each sample (Appendix A), which corresponds to the minimal coverage recommended by ENCODE (https://www.encodeproject.org/about/experiment-guidelines/). Read quality was assessed with FastQC (https://www.bioinformatics.babraham.ac.uk/projects/fastqc/), adapter sequences were removed with TrimGalore! v0.6.4 (http://www.bioinformatics.babraham.ac.uk/projects/trim_galore/) and mapped against the GRCz11 zebrafish reference genome using RNA-STAR v020201 [72] and the known exon–exon junctions from Ensembl release 95 [73]. Normalisation and differential expression analysis were performed with DESeq2 v1.22.2 [74]. Biological reproducibility was assessed by hierarchical clustering of the variance stabilised expression data (rlog) obtained from DESeq2 with Pearson’s correlation and complete linkage method. To increase the power of analysis of the differential gene expression analysis, we selected 37 samples with good reproducibility from the original set of 45 samples (Appendix A). Genes with |fold change| ≥ 1.5 and adjusted *p*-value < 0.01 (false discovery rate) were considered as differentially expressed in all analysis, except for the comparison of RNA-seq data with DMR where a threshold for the adjusted *p*-value < 0.05 was used. Hierarchical clustering of normalised expression data (rlog) or fold changes from DESeq2 were made with the R package hclust using Pearson’s correlation and the complete-linkage method. Fuzzy-mean clustering was used to cluster genes based on their expression patterns across the five different dose rates of ionising radiation. To do so, we normalised on the same scale, the fold change of 3319 significant DEG (|fold change| ≥ 1.5 and adjusted *p*-value < 0.01) in at least one condition and applied fuzzy-mean clustering [75] using the R package mfuzz using the parameters c = 4 and m = 2. Expression profiles were plotted with the R package ggplot2.

### 4.4. Gene Ontology Analysis

Gene ontology (GO) enrichment was performed using the R packages topGO [76] and clusterProfiler [77] using the zebrafish DEG obtained from DESeq2 (Appendix A). For KEGG enrichment, the human orthologous genes were used, as described before [25]. Enrichments with *p*-value from Fisher’s exact test ≤ 0.01 were considered significant. MA-plots, heat maps, histograms and Venn diagrams were produced using the R package ggplot2 [78].

### 4.5. TF-Binding Sites Enrichment

Significantly (|fold change| ≥ 1.5 and adjusted *p*-value < 0.01) DEG found at 50, 5 and 0.5 mGy/h were searched for enrichment of TF DNA-binding sites in their promoter sequence. A set of 5000 randomly chosen promoters was used as background. Promoter sequences were defined as DNA genomic sequence located 2 kb upstream and 50 bp downstream the transcriptional start site, and were retrieved with the Perl Applied Program Interface from Ensembl release 95. The resulting promoter sequences were searched for enrichment of TFBS with oPOSSUM v3 [79] using the hidden Markov matrix models from Jaspar core-vertebrate database v2018 [80]. Binding sites with z-score > 6 and Fisher-score > 3 were considered significantly enriched.

### 4.6. Purification of Genomic DNA

Biological replicates (3 per condition) were made from pools of 40 embryos at 6 hpf. Embryos were incubated under agitation at 55 °C during 3 h in extraction buffer (Tris 80 mM, NaCl 200 mM, EDTA 5 mM and SDS 0.5%) with 1 mg/mL of proteinase K. To avoid RNA contamination, samples were incubated 30 min at 37 °C with RNAse A (1 mg/mL). A phenol/chloroform extraction was performed followed by DNA precipitation with isopropyl alcohol. DNA pellets were resuspended in buffer AE (Qiagen, Courtaboeuf, France) and concentration assessed by fluorometric quantification (Qubit, Life Technologies, Villebon-sur-Yvette, France). Quality of DNA was assessed by electrophoresis on 1% (w/v) agarose gel; no signs of DNA degradation or RNA contamination were detected.

### 4.7. Bisulphite Conversion and Production of WGBS Data

Precisely, 300 ng of extracted genomic DNA was used for bisulphite conversion using the EZ DNA Methylation-Gold Kit (Zymo research, Irvine, USA) following manufacturer’s instructions. All samples were spiked-in with 0.5 ng of PUC 19 nonmethylated DNA (Zymo research, Irvine, USA) to control bisulphite conversion rate (Appendix A). Quality control of converted DNA was assessed on RNA Pico Chip (Bioanalyzer 2011, Agilent, Les Ulis, France) and by measuring recovered ssDNA quantity with a Nanodrop (Thermo Fisher, Villebon-sur-Yvette) spectrophotometer. Sequencing libraries were prepared from 50 ng of converted DNA with TruSeq DNA Methylation Kit and indexes from Illumina following manufacturer’s instructions. DNA integrity, quality and concentration were assessed on High Sensitivity DNA Chip (Bioanalyzer 2011, Agilent, Les Ulis, France). Libraries were multiplexed at 2 nM and run on a HiSeq 4000 (Illumina, San Diego, USA) to produce 50 bases long paired-end reads at the CerBMgie platform (IGBMC, Illkirch, France).

### 4.8. Bioinformatic Analysis of WGBS Data

Between 154 and 181 million of good-quality reads were produced for each sample (Appendix A). Read quality was assessed with FastQC (https://www.bioinformatics.babraham.ac.uk/projects/fastqc/). Adapter sequences and low read quality (Phred score < 30) were trimmed with TrimGalore! v0.6.4 (http://www.bioinformatics.babraham.ac.uk/projects/trim_galore/). Trimmed reads were mapped against a fully C-to-T converted version of the zebrafish genome (GRCz11) with Bismark v0.16.3 [81] with the following options: --bowtie2 -X 1000 -N 1 --ambig_bam --nucleotide_coverage -un –ambiguous --phred33-quals. Methylation bias (M-bias) plots were generated with bismark_methylation_extractor with the options --mbias_only --no_overlap. PCR duplicates were removed using the Bismark function deduplicate_bismark and methylation levels obtained with bismark_methylation_extractor using the following options: --bedGraph --cytosine_report --gzip -- --no_overlap -p --ignore_3prime 6 --ignore 13 --ignore_r2 13 --ignore_3prime_r2 6. Bisulphite conversion rates were >96% (Appendix A) and mapping efficiency against the zebrafish genome were >67% for all samples. Methylation levels in the CpG context were analysed with the R package DSS [82]. Methylation levels at each CpG site was determined in the biological triplicates from nearby CpGs using DMLtest function (window size of 300 base pairs) which uses an empirical Bayesian procedure to estimate the dispersion among all CpGs within the smoothing window. Methylation difference in pairwise comparison (exposed compared to control) was then assessed by a Wald test and adjusted *p*-value (False Discovery Rate) computed at each CpG site. Differentially methylated cytosines were considered significant at a threshold FDR < 5% and methylation differences ≥ 10%. Differentially methylated regions (DMR) were analysed with DMRichR [83,84,85] as a wrapper for dmrseq [84] and bsseq [85]. Briefly, CpG count matrix obtained from Bismark cytosine reports were filtered for at least 1X coverage per CpG across all samples. DMR were identified from pairwise comparison by a permutation test. To do so, each region with at least 3 CpG was compared to a null distribution created from a set of background regions meeting the same criteria (1X coverage across all samples and at least 3 CpG). Permutation *p*-value < 0.05 was considered significant, as published before [83]. Visualisation of DMR smoothed methylation values were generated with bsseq.

### 4.9. Association with Genomic Features

Heatmap of DEG associated with germ layers information was obtained by querying the expression database from ZFIN for the stage “gastrula” and the expression domains “ectoderm,” “endoderm” and “mesoderm” and selecting the DEG with |fold change| ≥ 1.5 and adjusted *p*-value < 0.01 in at least one comparative analysis. Clustering of expression differences (fold change) data was performed using the default of the R package heatmap.2. For CpG analysis, the genomic coordinates of known TSS were retrieved from all Ensembl transcripts (GRCv11 release 95) via the R package GenomicFeatures and GenomicRanges [86]. CpG islands, LINE, SINE and transposon genomic coordinates were retrieved from the USCS database (danRer11). Dot plot of cytosine methylation levels mapped to TSS and CpG islands were produced with the R packages EnrichedHeatmap [87] and circlize [88] using a distance to the feature of interest of 2 kb and a sliding window of 50 bp. Mapping of DMR coordinates to genomic features was performed with the R packages GenomicRanges and ChIPSeeker [89]. Circos plot of DMR and DEG was produced with the R package circlize with DMR located <3 kb from the TSS.

### 4.10. Analysis of H3K3me3 and H3K27me3 ChIP Data

ChIP-seq datasets made on shield zebrafish embryos against H3K3me3 (SRR372771) and H3K27me3 (SRR372772) marks were trimmed for quality (Q > 10). Adapter removal was made with TrimGalore! v0.6.4. Reads were then mapped against the zebrafish genome (GRCv11) with bowtie v1.2.2 using the parameter -m1. Uniquely mapped reads were used for peak detection using MACS2 using an effective genome size of 1.3.10^9^ bp. Genomic coordinates of the ChIP peaks were manipulated in R with the package GenomicRanges and dot plot of cytosine methylation levels mapped around H3K27me3 and H3K4me3 produced with the R packages EnrichedHeatmap and circlize.

## 5. Declarations

### 5.1. Ethics Approval and Consent to Participate

Animals were housed in the IRSN animal facilities accredited by the French Ministry of Agriculture for performing experiments on live zebrafish. Animal experiments were performed in compliance with French and European regulations on protection of animals used for scientific purposes (EC Directive 2010/63/EU and French Decree 2013-118). All experiments were approved by the Ethics Committee #81 and authorised by the French Ministry of Research under the reference APAFIS#11488.

### 5.2. Availability of Data and Materials

The datasets generated in this study are publicly available in the GEO repository under the accession number GSE146198.

## Figures and Tables

**Figure 1 ijms-21-04014-f001:**
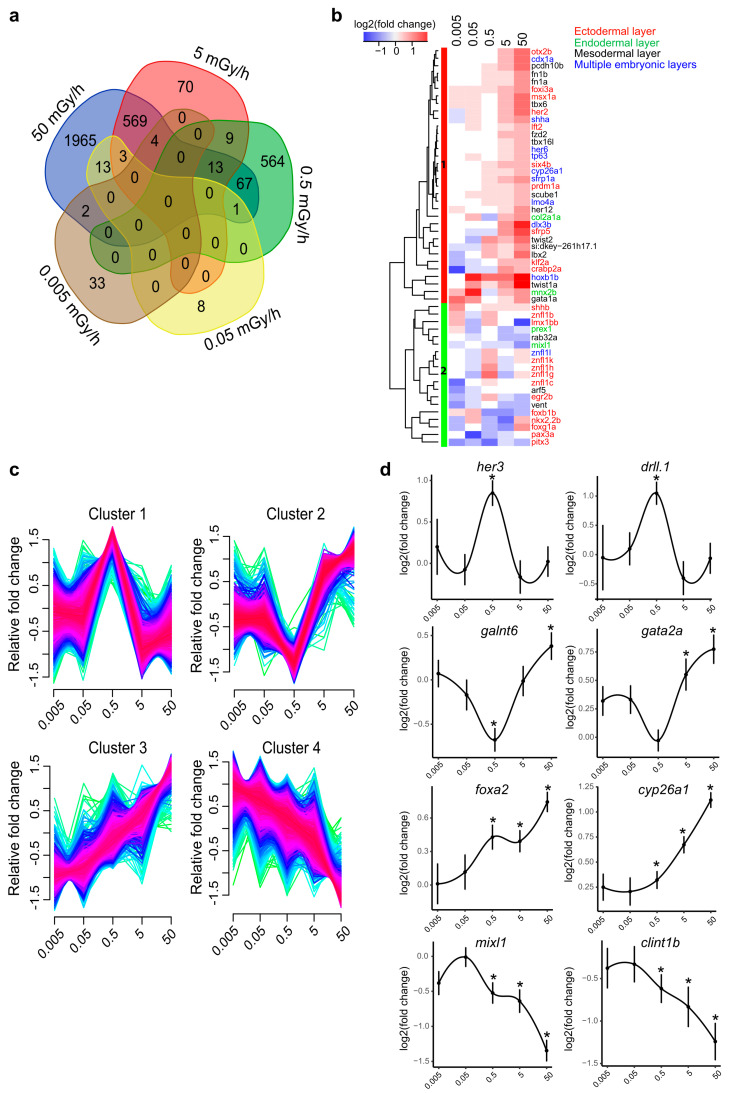
Transcriptomic analysis of shield embryos (6 hpf) exposed to ionising radiation (IR). Dose rates are indicated in mGy/h. (**a**) Venn diagram of differentially expressed genes at the five dose rates (|fold change| ≥ 1.5 and adjusted *p*-value < 0.01). (**b**) Hierarchical clustering of log2 fold change across all dose rates of genes known to be expressed in the different germ layers during gastrulation (based on ZFIN annotations, see Material and Methods). Known expression in the ectoderm (red), endoderm (green), mesoderm (black) or in multiple embryonic layers (blue) is indicated. Upregulated genes are displayed in red, downregulated genes in blue and no changes in white. Cluster 1 (red) and cluster 2 (green) are indicated. (**c**) Expression patterns obtained by fuzz-mean clustering of 3319 selected DEG misregulated in at least one condition (|fold change| ≥ 1.5 and adjusted *p*-value < 0.01). The fold-change were normalised on the same scale, and the y-axes indicate relative fold change. Genes with high cluster membership are displayed as red lines, those with moderate membership in blue and low membership in green. (**d**) Example of gene expression patterns across the different dose rates using loess-smoothed conditional means. Two genes are displayed for each of the four clusters detected by fuzzy-mean clustering. Mean of log2 (fold change) are indicated as dots and standard errors are indicated as vertical bars (* adjusted *p*-value < 0.01).

**Figure 2 ijms-21-04014-f002:**
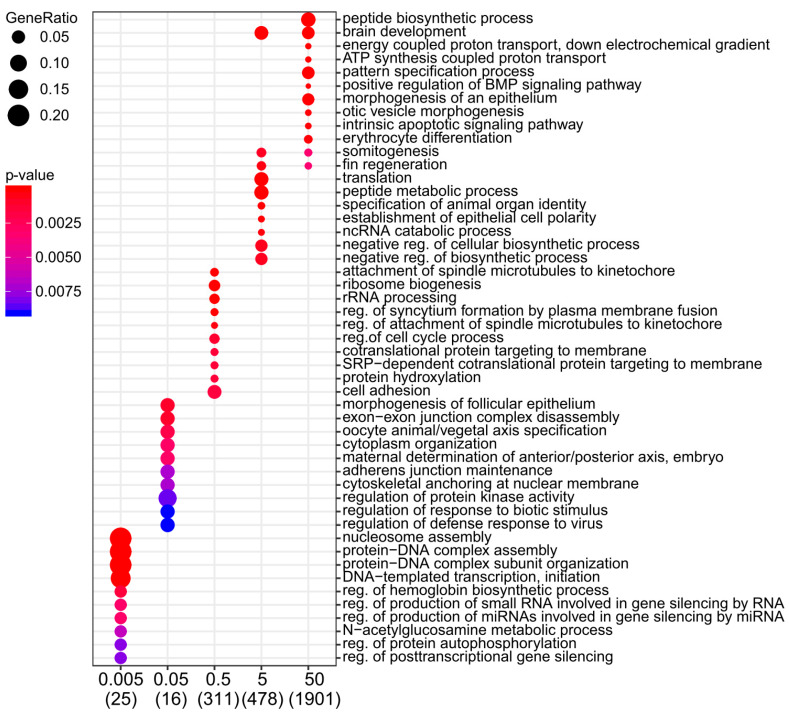
Dot plot of zebrafish GO (Gene Ontology) term enrichment showing the top 10 enriched pathways. Dose rates are indicated at the bottom in mGy/h. The total number of deregulated genes within the GO pathways selected on the dot plot are indicated in brackets. Colours indicate the *p*-values from Fisher’s exact test, and dots size is proportional to the number of differentially expressed genes (DEG) in the given pathway.

**Figure 3 ijms-21-04014-f003:**
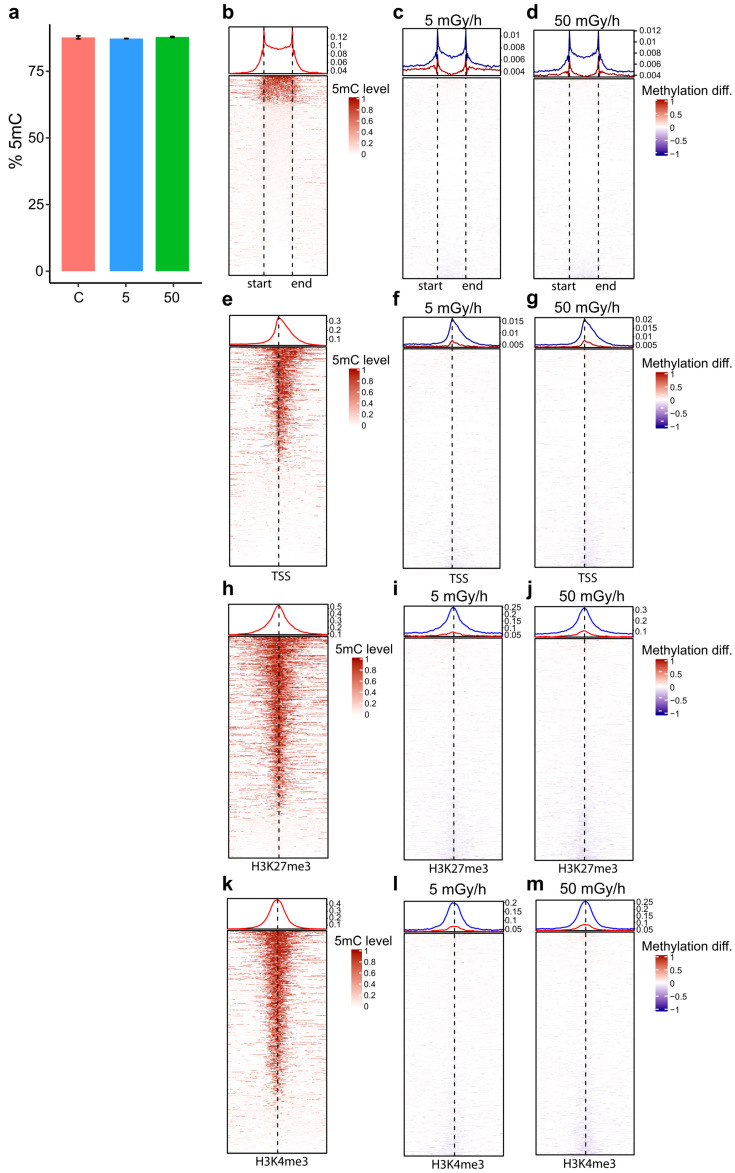
Analysis of 5mC abundancy in the CpG context after exposure to IR at 5 and 50 mGy/h in shield embryos. (**a**) Mean percentage of 5mC in control (C, red), 50 mGy/h (50, green) and 5 mGy/h (5, blue). Standard errors for the biological replicates are indicated. (**b**–**m**) Dot plots of 5mC mapped in a window 2 kb up and down to known genomic features. Start–end of CpG islands (CGIs) as well as localisation of transcriptional start (TSS) or the histone marks H3K27me3 and H3K4me3 are indicated by the dashed lines. The fraction of 5mC in the +/− 2 kb genomic window is indicated at the top of each dot plot. (**b**) Fraction of 5mC in control embryos in CGIs, (**e**) in TSS, (**h**) in H3K27me3 and (**k**) in H3K4me3. Fraction of differentially methylated cytosine (5mC) in exposed embryos (5 and 50 mGy/h) in (**c** and **d**) CGIs, (**f** and **g**) TSS, (**i** and **j**) H3K27me3 and (**l** and **m**) H3K4me3. Blue: hypomethylated compared to control and red: hypermethylated.

**Figure 4 ijms-21-04014-f004:**
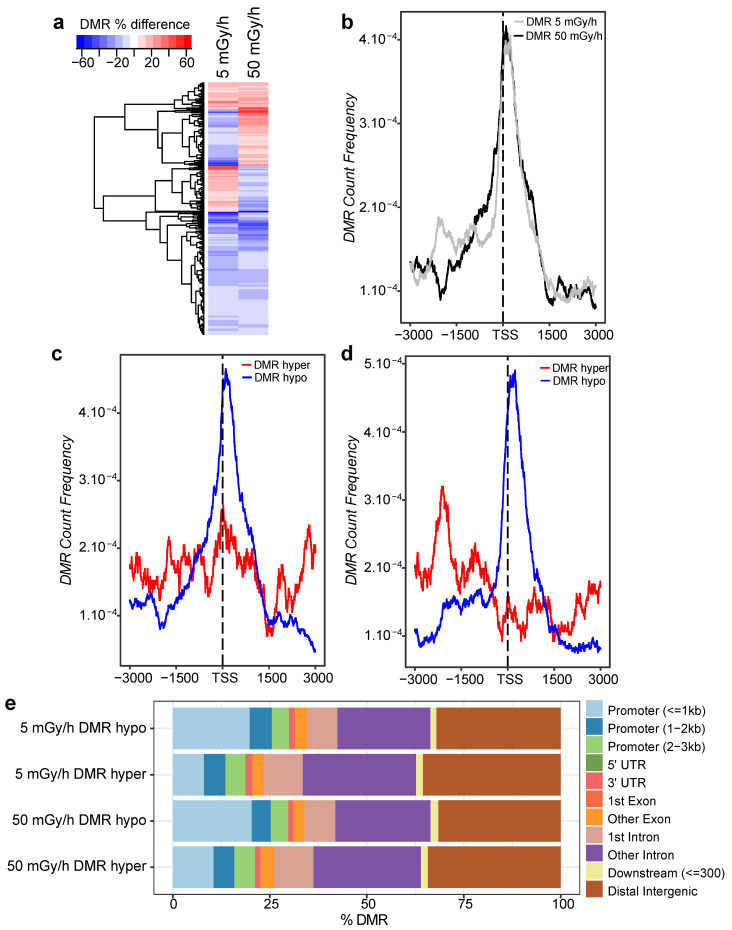
Analysis of differentially methylated regions (DMR). (**a**) Heatmap of methylation difference (%) in DMR detected at 5 and 50 mGy/h. Blue DMR: hypomethylation, red DMR: hypermethylation and white: no change. (**b**) DMR count frequency in a 6 kb window relative to known TSS (dashed line) at 50 mGy/h (black) and 5 mGy/h (grey). (**c**) Distribution of hypomethylated (blue) and hypermethylated (red) DMR in a 6 kb window centred to known TSS (dashed line) at 50 mGy/h and (**d**) at 5 mGy/h. (**e**) Distance of hypermethylated and hypomethylated DMR to known genomic features (TSS, gene promoter, exons, introns and intergenic regions).

**Figure 5 ijms-21-04014-f005:**
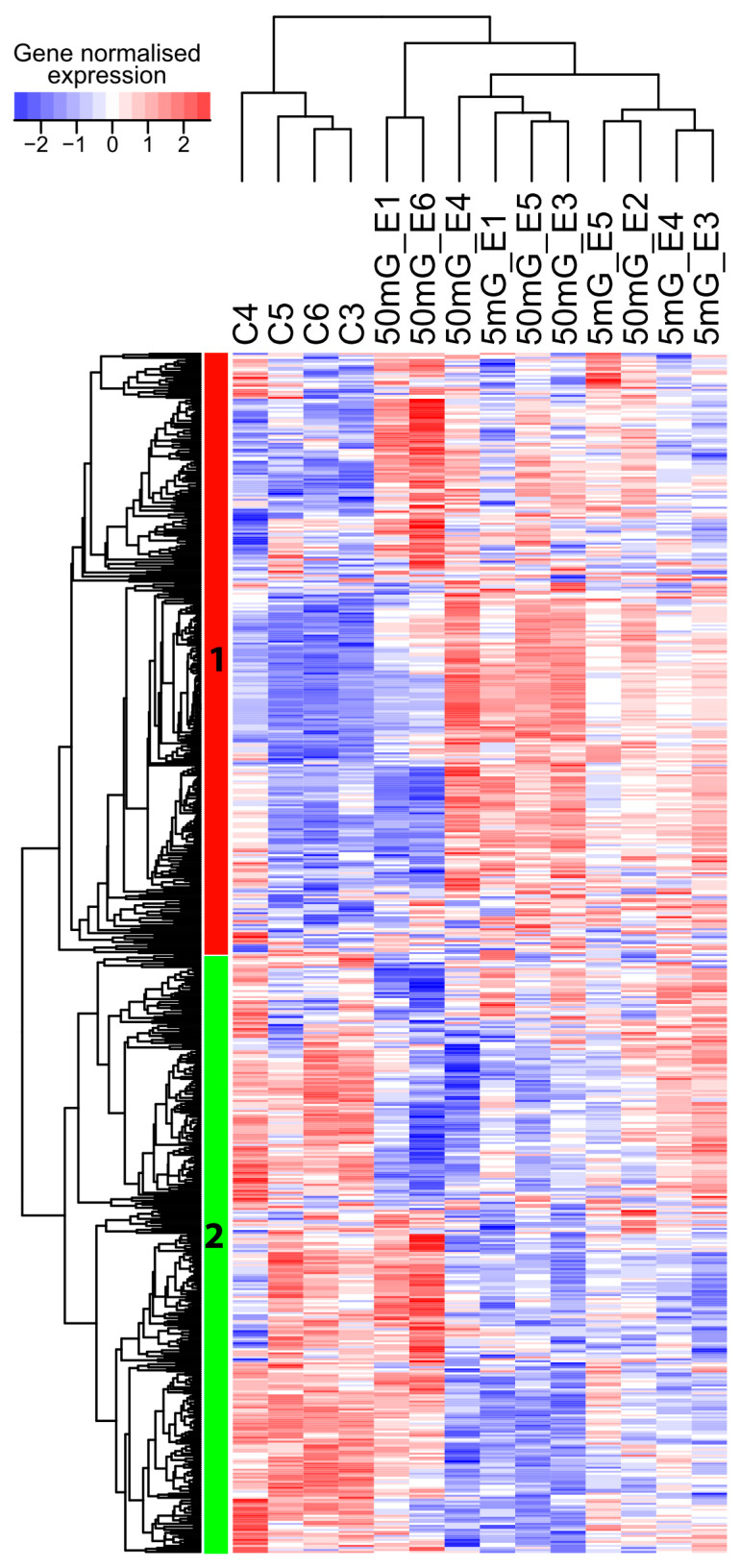
Heat map of normalised expression of 600 genes with a hypomethylated promoter (DMR < 500 bp from TSS) at 50 mGy/h (50 mG), 5 mGy/h (5 mG) and control (C). Biological replicates in each condition are displayed. Blue: low expression, white: moderate expression and red: high expression.

**Figure 6 ijms-21-04014-f006:**
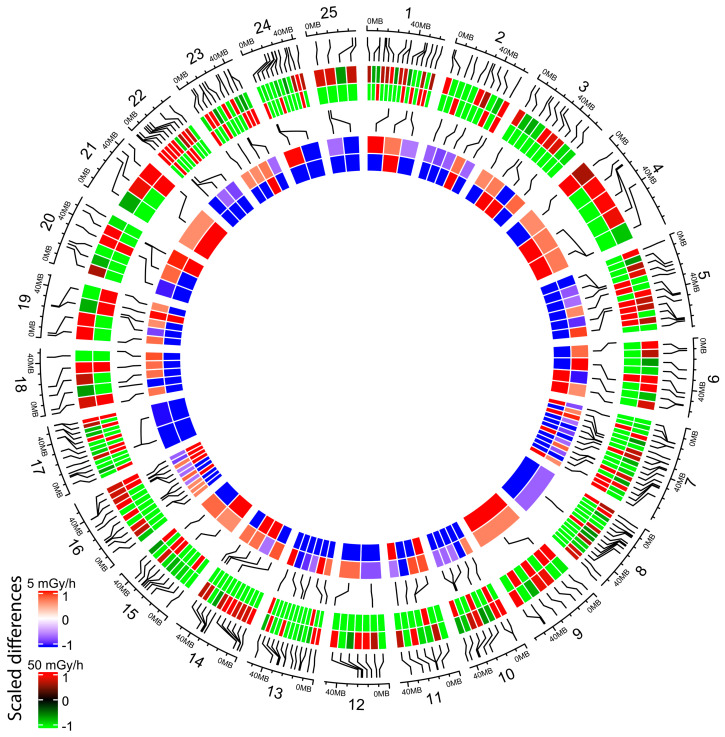
Circos plot of differentially methylated promoter associated with DEG at 50 mGy/h (two outer circles) and 5 mGy/h (two inner circles). Zebrafish chromosome number are indicated from 1 to 25 and approximate genomic locations are displayed by black lines. Promoter methylation changes (defined as DMR located at < 3 kb from TSS, % methylation difference ≥ 10% and permutation *p*-value < 0.05) and gene expression changes (fold change of DEG with adjusted *p*-value < 0.05) were normalised on the same scale to visualise effect of promoter methylation on gene expression. Outer circle: % methylation difference in DMR at 50 mGy/h (hypermethylation: red, no change: white and hypomethylation: green). Second circle: DEG at 50 mGy/h (upregulation: red, no change: white and downregulation: green). Third circle: % methylation difference in DMR at 5 mGy/h (hypermethylation: red, no change: white and hypomethylation: blue). Inner circle: DEG at 5 mGy/h (upregulation: red, no change: white and downregulation: blue).

**Figure 7 ijms-21-04014-f007:**
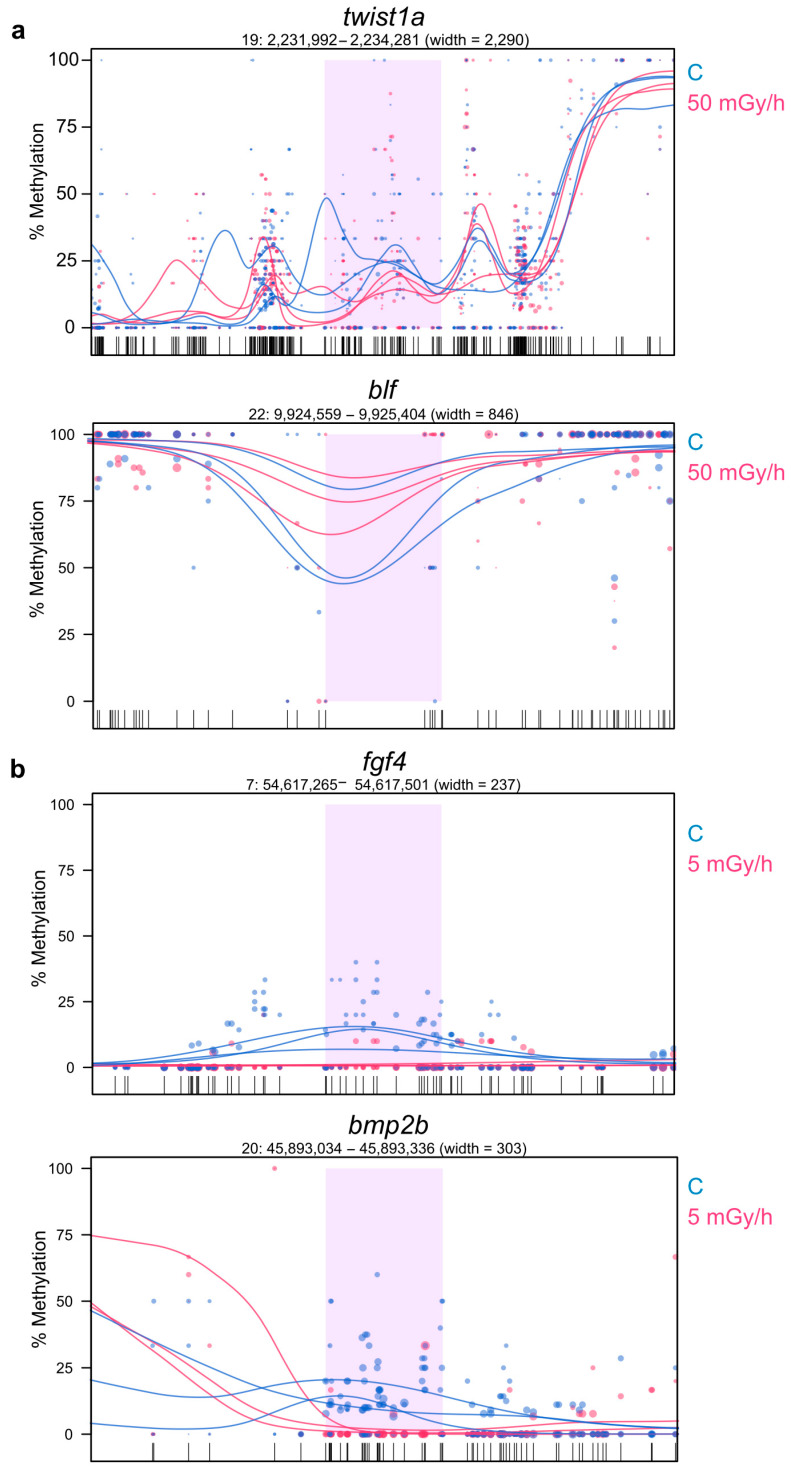
Example of methylation profile of DMR located in the promoter of DEG gene at (**a**) 50 mGy/h and (**b**) 5 mGy/h. The purple part corresponds to the DMR detected with significant methylation change (permutation *p*-value < 0.05 and methylation difference ≥ 10%). The gene name and the genomic coordinated are indicated at the top. Locations of methylated CpG measured in the whole genome bisulphite sequencing (WGBS) data are indicated at the bottom by black vertical lines. Methylation of CpG appears as circles for each biological replicate with a dot size proportional to the reads coverage. The lines correspond to the model of CpG methylation level for each sample. Red: 50 or 5 mGy/h and blue: control.

**Figure 8 ijms-21-04014-f008:**
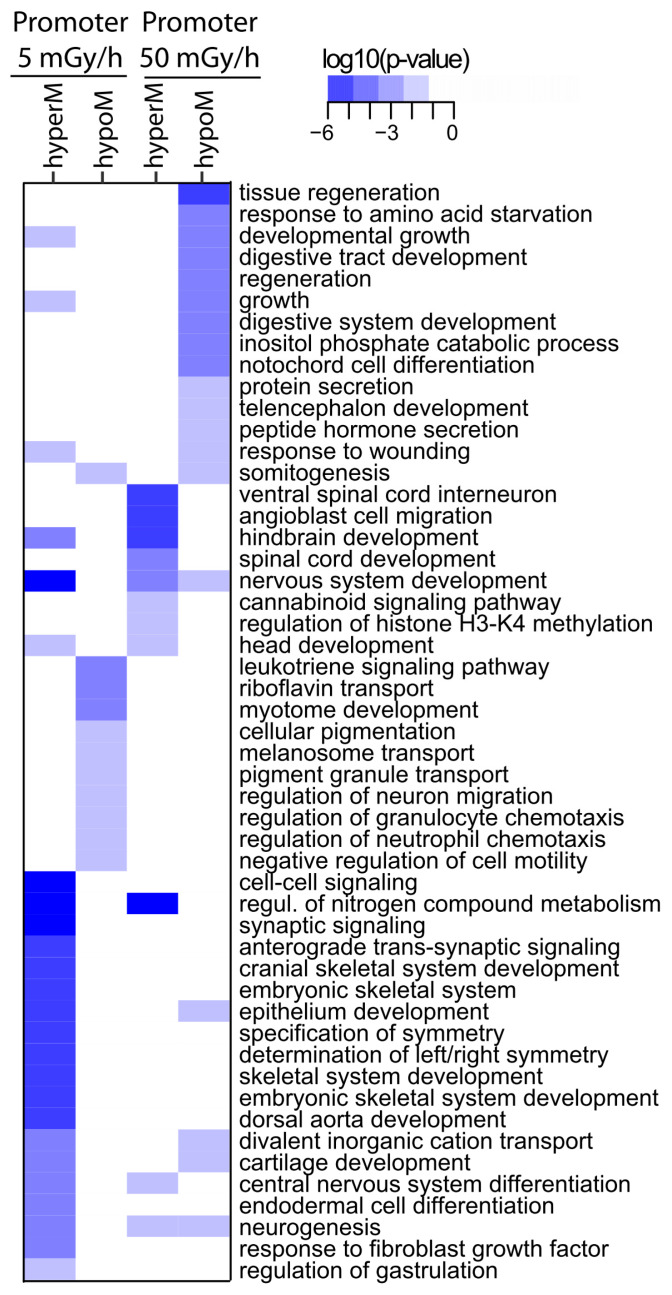
Heat map of GO enrichment of gene promoter displaying significant methylation changes. DMR with permutation *p*-values < 0.05 and located less than 3 kb from the TSS were used. Dose rates and methylation state are indicated at the top. Colours indicate the enrichment *p*-values obtained from Fisher’s exact test.

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
