# Peer review of "Ionising Radiation Induces Promoter DNA Hypomethylation and Perturbs Transcriptional Activity of Genes Involved in Morphogenesis during Gastrulation in Zebrafish"

_ijms, 2020, doi:10.3390/ijms21114014_

Round 1

Reviewer 1 Report

The authors should consider the followings:

  1. In Fig 1b, the authors should give rationale of why many of the genes did not show a dose-dependent response (of radiation), to their fold-changes.
  2. Did the authors measure any irregularity of the zebrafish, such as developmental delay, for those embryos that undergone the range of radiation?
  3. Were those observation reported in this article in only at the transcriptional or epigenetic level? did the authors study any protein level variation for those irradiation treated groups? Did the protein changes agree with the upstream changes?
  4. In methodology, on “0.0044 mGy/h ± 0.0001 mGy/h”, please give rationale, with the lowest dosage within a reliable range of measurement uncertainty.
  5. Why did the author choose, Fold changes of >=1.5, instead of a higher Fold-changes, like >=2? Would switching to FC of 2, giving a great differences of the results/ subsequent analyses?
  6. Were the findings comparable to any radiation studies on mouse embryos (or other mammalian)?
  7. The authors should give information of any traceable evidences of toxicity of the zebrafish embryos treated with the range of radiation.
  8. In Figure 1a, please explain the rationale of the varied number of differential changes solely upon the increasing mGy/h. e. Why are there 33 (solely) differential changes of genes on 0.005 mGy/h, while increasing to 0.05 mGy/h it dropped to 8, further increasing to 0.5 mGy/h, it increased further to 564 (solely) genes with differential changes; then it dropped to 70 at 5 mGy/h?
  9. The authors may later couple their studies with KO/KI study, with functional assays and changes in critical protein levels, for further validation.
  10. Did the authors assure no earlier exposure of the radiation (before 1hpf)? how did the author verfiy that?
  11. Please give rationale(s) in the article, for how the authors decide the sequencing depth.
  12. Did the author report any differential changes in lncRNA (after treatment)?
  13. Please state clearly the novelty of this research in your abstract and conclusion.
  14. The authors should consider using English proof-reading services by language professional.

Author Response

  1. In Fig 1b, the authors should give rationale of why many of the genes did not show a dose-dependent response (of radiation), to their fold-changes.

We want to thank the reviewer for this very interesting question. In the figure 1b, most of the genes in cluster 1 display a higher upregulation at 50 mGy/h than 5mGy/h and 0.5 mGy/h. At lower dose rates the changes are either non-significant or non-linear to the doses. To further investigate this point we analysed the expression patterns of 3319 DEG (misregulated in at least one condition) by fuzzy-mean clustering. We found 4 main clusters of genes, two of them followed a dose-dependent response to radiations. These results suggest that a linear dose response to irradiation is not a common feature to all genes and all dose rates. Rather, we observe that specific set of genes respond linearly to the dose between 50 mGy/h and 0.5 mGy/h, but not at lower dose rate. We added the panel Fig1c and 1d to illustrate this point. We also added a paragraph in the results lines 118 to 126 and in the discussion line 262 to 268. We also added the corresponding material and methods and the reference for the usage of fuzzy mean clustering. A comment on known non-linear dose response to radiation was also added line 26 with the corresponding reference.

  1. Did the authors measure any irregularity of the zebrafish, such as developmental delay, for those embryos that undergone the range of radiation?

No morphological abnormalities or increased embryonic lethality could be observed at any of the dose rates tested here. This is in agreement with earlier studies that showed that exposures to IR at < 150 mGy does not impact embryonic survival directly (Bladen 2005), but rather induced subtle neuromuscular and motility defects (S. Murat 2019). A sentence was added in the results line 101 to 102 and in the discussion line 252 to 255.

  1. Were those observation reported in this article in only at the transcriptional or epigenetic level? did the authors study any protein level variation for those irradiation treated groups? Did the protein changes agree with the upstream changes?

The present manuscript describes the molecular effects at the transcriptomic and DNA methylation levels. We didn’t produce protein analysis on gastrula stage. This was done only on 96hpf larvae, and the data were published in our precedent paper (S. Murat et al. 2019). We demonstrated that similar biological pathways were induced at the transcriptomics and proteomic scale. However, we also showed that a direct comparison of gene expression at the transcript level and at protein level is usually not possible. Such observation is in accordance with other studies that described < 10% of misregulated genes have similar expression changes at the protein level. We would like to point to this reviewer that the present manuscript focuses on the global analysis at the transcriptomics and epigenetics levels. Proteomics or western-blot of particular candidates could be interesting for future functional studies, but this was not the aim in the present manuscript.

  1. In methodology, on “0.0044 mGy/h ± 0.0001 mGy/h”, please give rationale, with the lowest dosage within a reliable range of measurement uncertainty.

We thank the reviewer for this remark. The standard deviation was computed on this table with the corrected dose rates obtained from six different MCNP simulations. We now rounded the standard deviation to only 2 decimals in the Supplementary table T4, but for clarity we did not change these numbers in the Material and Methods.

  1. Why did the author choose, Fold changes of >=1.5, instead of a higher Fold-changes, like >=2? Would switching to FC of 2, giving a great differences of the results/ subsequent analyses?

There is no consensus in the literature about a single value for the fold change or the adjusted p-values. Classical values for the fold change are in the range of 1.5 to 2 and for the adjusted p-values between 0.01 and 0.05. In this manuscript we have chosen a rather stringent adjusted p-value (< 0.01) and a relaxed criteria for the fold change (>1.5), because we believe that adjusted p-value is a better criteria to detect false positives than fold change. Increasing the threshold of the fold change to 2 has an effect on the total number of misregulated genes, but does not have strong impacts on the enriched biological processes. For instance, increasing the FC to 2 results in 1034 DEG and the GO analysis again highlights enrichment of pathways like Notch signaling, somitogenesis, epiboly and dorsal/ventral pattern formation. We thus think that our criteria for DEG selection are valid and in the range of what is currently found in the literature.

  1. Were the findings comparable to any radiation studies on mouse embryos (or other mammalian)?

The vast majority of the mouse studies focus on acute exposure to X-rays during organogenesis (E9-E15 in the mouse). In the introduction we referred to Bernal et al. FASEB J. 27, 665–71 (2013) who studied DNA methylation at the Avy locus at 0.7 cGy. We are not aware of any mouse studies focusing on chronic irradiation to gamma radiation on gastrula embryos (near E6.5 in the mouse). However, we cite in the discussion human epidemiological studies that showed that low dose of radiation have neurological effects. The comparison between mammalian data is thus very indirect.

  1. The authors should give information of any traceable evidences of toxicity of the zebrafish embryos treated with the range of radiation.

This point is similar to the question 2 asked by the same reviewer. We did not find overt sign of embryonic death or morphological problems in the dose rates tested here. This observation is in good agreement with the published literature. Our data are focused on the molecular effects, and we argue that this molecular changes are predictive of the neuromuscular defects observed later in development.

  1. In Figure 1a, please explain the rationale of the varied number of differential changes solely upon the increasing mGy/h. e. Why are there 33 (solely) differential changes of genes on 0.005 mGy/h, while increasing to 0.05 mGy/h it dropped to 8, further increasing to 0.5 mGy/h, it increased further to 564 (solely) genes with differential changes; then it dropped to 70 at 5 mGy/h?

We added a sentence in the discussion line 258 to 261 about the high number of unique genes at 50 mGy/h and 0.5 mGy/h, which could underlie (in part) a specific transcriptional response at these dose rates. However, and in contrast to the suggestion made by this reviewer, we do not believe that the number of unique genes in each condition can be compared directly as it is the results of many factors, including the total number of DEG and the overlap with other conditions. In our point of view the main points are the number of total DEG which increase with the dose rate, and the observation that 50 mGy/h and 5 mGy/h have many genes in common.

  1. The authors may later couple their studies with KO/KI study, with functional assays and changes in critical protein levels, for further validation.

We agree with the reviewer and added a sentence line 352 to 353.

  1. Did the authors assure no earlier exposure of the radiation (before 1hpf)? how did the author verfiy that?

Adult and freshly layed eggs were not exposed before irradiation but subjected to the background level of radiation which is found everywhere. The operational measure of the background level showed that it was always < 0.1 µGy/h (0.0001 mGy/h). We added this information in the Material and Methods line 448.

  1. Please give rationale(s) in the article, for how the authors decide the sequencing depth.

We followed the ENCODE’s recommendations: minimal coverage between 20M and 30M reads for DEG analysis using RNAseq data. We added a sentence in the manuscript line 456

  1. Did the author report any differential changes in lncRNA (after treatment)?

Despite interesting, lncRNAs were not investigated in our current analysis, but such analysis might be made in future studies using dedicated pipeline and analysis.

  1. Please state clearly the novelty of this research in your abstract and conclusion.

We modified two sentences in the abstract, line 26 to 28.

  1. The authors should consider using English proof-reading services by language professional.

Minor English errors were corrected in the whole manuscript.

Reviewer 2 Report

The Authors have investigated an interesting topic and the theme has been properly described and novel findings have been reported. I would like to congratulate Authors for the good-quality of the article, the literature reported used to write the paper, and for the clear and appropriate structure. The manuscript is well written, presented and discussed, and understandable to a specialist readership.

In general, the organization and the structure of the article are satisfactory and in agreement with the journal instructions for authors. The subject is adequate with the overall journal scope. The work shows a conscientious study in which a very exhaustive discussion of the literature available has been carried out. The introduction provides sufficient background, and the other sections include results clearly presented and analyzed exhaustively.

So, I recommend the acceptance of the paper in Int. J. Mol. Sci.

Author Response

Thank you for your nice comments.